# Mitigating Interference in the Knowledge Continuum through Attention-Guided Incremental Learning

## Abstract

Continual learning (CL) remains a significant challenge for deep neural networks, as it is prone to forgetting previously acquired knowledge. Several approaches have been proposed in the literature, such as experience rehearsal, regularization, and parameter isolation, to address this problem. Although almost zero forgetting can be achieved in task-incremental learning, class-incremental learning remains highly challenging due to the problem of inter-task class separation. Limited access to previous task data makes it difficult to discriminate between classes of current and previous tasks. To address this issue, we propose 'Attention-Guided Incremental Learning' (AGILE), a novel rehearsal-based CL approach that incorporates compact task-attention to effectively reduce interference between tasks. AGILE utilizes lightweight, learnable task projection vectors to transform the latent representations of a shared task-attention module toward task distribution. Through extensive empirical evaluation, we show that AGILE significantly improves generalization performance by mitigating task interference and outperforms rehearsal-based approaches in several CL scenarios. Furthermore AGILE can scale well to a large number of tasks with minimal overhead while remaining well-calibrated with reduced task-recency bias[2].

## 1 Introduction

In recent years, deep neural networks (DNNs) have been shown to perform better than humans on certain specific tasks, such as Atari games (Silver et al., 2018) and classification (He et al., 2015). Although impressive, these models are trained on static data and are unable to adapt their behavior to novel tasks while maintaining performance on previous tasks when the data evolves over time (Fedus et al., 2020). Continual learning (CL) refers to a training paradigm in which DNNs are exposed to a sequence of tasks and are expected to learn potentially in an incremental or online manner (Parisi et al., 2019). CL has remained one of the most daunting tasks for DNNs, as acquiring new information significantly deteriorates the performance of previously learned tasks, a phenomenon termed "catastrophic forgetting" (French, 1999; McCloskey & Cohen, 1989). Catastrophic forgetting arises due to the stability-plasticity dilemma (Mermillod et al., 2013), the degree to which the system must be stable to retain consolidated knowledge while also being plastic to assimilate new information. Catastrophic forgetting often results in a significant decrease in performance, and in some cases, previously learned information is completely erased by new information (Parisi et al., 2019).

Several approaches have been proposed in the literature to address the problem of catastrophic forgetting in CL. Rehearsal-based approaches (Ratcliff, 1990) explicitly store a subset of samples from previous tasks in the memory buffer and replay them alongside current task samples to combat forgetting. In scenarios where the buffer size is limited due to memory constraints (e.g., edge devices), these approaches are prone to overfitting on the buffered data (Bhat et al., 2022). On the other hand, regularization-based approaches (Kirkpatrick et al., 2017) introduce a regularization term in the optimization objective and impose a penalty on changes in parameters important for previous tasks. Although regularization greatly improves stability, these approaches cannot discriminate classes from

---

[2]Code will be made publicly available upon acceptance.

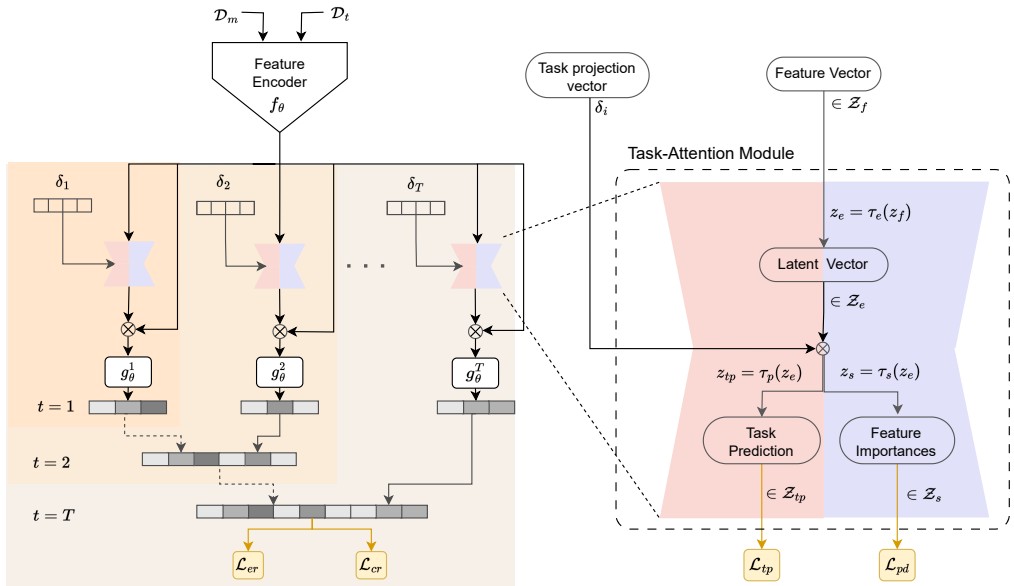

Figure 1: **A**ttention-**G**uided **I**ncremental **Le**arning (AGILE) consists of a shared task-attention module and a set of task-specific projection vectors, one for each task. Each sample is passed through the task-attention module once for each projection vector, and the outputs are fed into task-specific classifiers. AGILE effectively reduces task interference and facilitates accurate task-id prediction (TP) and within-task prediction (WP).

different tasks, thus failing miserably in scenarios such as Class-Incremental Learning (Class-IL) (Lesort et al., 2019). Parameter isolation approaches limit interference between tasks by allocating a different set of parameters for each task, either within a fixed model capacity (Gurbuz & Dovrolis, 2022) or by expanding the model size (Rusu et al., 2016). However, these approaches suffer from several shortcomings, including capacity saturation and scalability issues in longer task sequences. With an increasing number of tasks, selecting the right expert in the absence of task identity is nontrivial (Aljundi et al., 2017), and therefore limits their application largely to Task-Incremental Learning (Task-IL).

The problem of inter-task class separation in Class-IL remains a significant challenge due to the difficulty in establishing clear boundaries between classes of current and previous tasks (Lesort et al., 2019). When a limited number of samples from previous tasks are available in the buffer in experience rehearsal, the CL model tends to overfit on the buffered samples and incorrectly approximates the class boundaries. Kim et al. (2022) decomposes the Class-IL problem into two sub-problems: task-id prediction (TP) and within-task prediction (WP). TP involves identifying the task of a given sample, while WP refers to making predictions for a sample within the classes of the task identified by TP. Therefore, the Class-IL problem can be seen as a combination of the Task-IL problem (WP) and the task discovery (TP). Regardless of whether the CL algorithm defines it explicitly or implicitly, good TP and good WP are necessary and sufficient to ensure good Class-IL performance (Kim et al., 2022). As task interference adversely affects both WP and TP, we hypothesize that focusing on the information relevant to the current task can facilitate more accurate TP and WP by filtering out extraneous or interfering information.

To this end, we propose 'Attention-Guided Incremental Learning' (AGILE), a rehearsal-based novel CL approach that encompasses compact task-attention to effectively mitigate interference between tasks and facilitate a good WP and TP in Class-IL. To further augment rehearsal-based learning in Class-IL, AGILE leverages parameter isolation to bring in task specificity with little computational or memory overhead. Specifically, AGILE entails a shared feature encoder and task-attention module, and as many task projection vectors as the number of tasks. Each task projection vector is a light-weight learnable vector associated with a particular task, specialized in transforming

the latent representations of shared task-attention module towards the task distribution. With dynamic expansion of task projection vectors, AGILE scales well to a large number of tasks while leaving a negligible memory footprint. Across CL scenarios, AGILE greatly reduces task interference and outperforms rehearsal-based approaches while being scalable and well-calibrated with less task-recency bias.

## 2 RELATED WORKS

**Rehearsal-based Approaches:** Earlier work sought to combat catastrophic forgetting in CL by explicitly storing and replaying previous task samples through Experience-Rehearsal (ER) (Ratcliff, 1990). Several works build on top of ER: Since soft targets carry more information and capture complex similarity patterns in the data compared to hard targets (Hinton et al., 2015), DER++ (Buzzega et al., 2020) enforces consistency in predictions through regularization of the function space. To further improve knowledge distillation through consistency regularization, CLS-ER (Arani et al., 2022) employs multiple semantic memories that better handle the stability-plasticity trade-off. More recent works focus on reducing representation drift right after task switching to mitigate forgetting: ER-ACE (Caccia et al., 2022) through asymmetric update rules shields learned representations from drastic adaptations while accommodating new information. Co$^2$L (Cha et al., 2021) employs contrastive representation learning to learn robust features that are less susceptible to catastrophic forgetting. However, under low-buffer regimes, these approaches are prone to overfitting. Under low-buffer regimes, the quality of the buffered samples plays a significant role in defining the ability of the CL model to approximate past behavior. GCR (Tiwari et al., 2022) proposed a core set selection mechanism that approximates the gradients of the data seen so far to select and update the memory buffer. In contrast, DRI (Wang et al., 2022a) employs a generative replay to augment the memory buffer under low buffer regimes. Although reasonably successful in many CL scenarios, rehearsal-based approaches lack task-specific parameters and run the risk of shared parameters being overwritten by later tasks.

**Task Attention:** As the weights in DNNs hold knowledge of previous tasks, intelligent segregation of weights per task is an attractive alternative to rehearsal to reduce catastrophic forgetting in CL. Dynamic sparse parameter isolation approaches (e.g., NISPA (Gurbuz & Dovrolis, 2022), CLNP (Golkar et al., 2019), PackNet (Mallya & Lazebnik, 2018)) leverage over-parameterization of DNNs and learn sparse architecture for each task within a fixed model capacity. However, these approaches suffer from capacity saturation and fail miserably in longer task sequences. By contrast, some parameter-isolation approaches grow in size, either naively or intelligently, to accommodate new tasks with the least forgetting. Progressive Neural Networks (PNN; (Rusu et al., 2016)) was one of the first works to propose a growing architecture with lateral connections to previously learned features to simultaneously reduce forgetting and enable forward transfer. Since PNN instantiates a new sub-network for each task, it quickly runs into scalability issues. Approaches such as CPG (Hung et al., 2019a) and PAE (Hung et al., 2019b) grow drastically slower than PNN, but require task identity at inference. HAT (Serra et al., 2018) employed a task-based layer-wise hard attention mechanism in fully connected or convolutional networks to reduce interference between tasks. However, layer-wise attention is quite cumbersome as many low-level features can be shared across tasks. Due to the limitations mentioned above, task-specific learning approaches have been largely limited to the Task-IL setting.

Although almost zero forgetting can be achieved in Task-IL (Serra et al., 2018), the Class-IL scenario still remains highly challenging due to the problem of inter-task class separation. Therefore, we propose AGILE, a rehearsal-based CL method that encompasses task attention to facilitate a good WP and TP by reducing interference between tasks.

## 3 PROPOSED METHOD

### 3.1 MOTIVATION

Task interference arises when multiple tasks share a common observation space but have different learning goals. In the presence of task interference, both WP and TP struggle to find the right class or task, resulting in reduced performance and higher cross-entropy loss. Continual learning in the

**Algorithm 1** Proposed Method: AGILE

1: **Input:** Data streams $\mathcal{D}_t$, Model $\Phi_\theta = \{f_\theta, \tau_\theta, \delta_\theta, g_\theta\}$, Hyperparameters $\alpha, \beta, \gamma, \lambda$, Memory buffer $\mathcal{D}_m \leftarrow \{\}$
2: **for all** tasks $t \in \{1, 2, .., T\}$ **do**
3:     **for all** epochs $e \in \{1, 2, .., E\}$ **do**
4:         Sample a minibatch $\{x_j, y_j\}_{j=1}^N \in \mathcal{D}_t$
5:         $\hat{y}_j, z_{sj}, z_{tpj} = \text{TASKATTENTION}(x_j)$
6:         $\mathcal{L} = \gamma\mathcal{L}_{tp} + \lambda\mathcal{L}_{pd}$
7:         **if** $\mathcal{D}_m \neq \emptyset$ **then**
8:             Sample a minibatch $\{x_k, y_k\}_{k=1}^N \in \mathcal{D}_m$
9:             $\hat{y}_k, z_{sk}, z_{tpk} = \text{TASKATTENTION}(x_k)$
10:         $\mathcal{L} \mathrel{+}= \mathcal{L}_{er} + \beta\mathcal{L}_{cr}$
11:         Update $\Phi_\theta$ and $\mathcal{D}_m$
12:         Update $\theta_{EMA}$
13: **Return:** model $\Phi_\theta$

**Algorithm 2** Task-Attention

**function** TASKATTENTION($x$):
    $z_f = f_\theta(x)$
    **for all** $i \leq t$ **do**
        $z_e^i = \tau^e(z_f)$
        $z_s^i = \tau^s(z_e^i \otimes \delta_i)$
        $z_{tp} = \tau^{tp}(z_e^i \otimes \delta_i)$
        $\hat{y}^i = g^i(z_s^i \otimes z_f)$
    $\hat{y}_j = concat(\hat{y}_j^i; \ \forall i \leq t)$
    **return** $\hat{y}, z_s, z_{tp}$

brain is governed by the conscious processing of multiple knowledge bases anchored by a rich set of neurophysiological processes (Goyal & Bengio, 2020). Global Workspace Theory (GWT) (Baars, 1994; 2005; Baars et al., 2021) provides a formal account of cognitive information access and posits that one such knowledge base is a common representation space of fixed capacity from which information is selected, maintained, and shared with the rest of the brain (Juliani et al., 2022). During information access, the attention mechanism creates a communication bottleneck between the representation space and the global workspace, and only behaviorally relevant information is admitted into the global workspace. Such conscious processing could help the brain achieve systematic generalization (Bengio, 2017) and deal with problems that could only be solved by multiple specialized modules (VanRullen & Kanai, 2021).

In functional terms, GWT as a model of cognitive access has several benefits for CL. (i) The common representation space is largely a shared function, resulting in maximum re-usability across tasks; (ii) The attention mechanism can be interpreted as a task-specific policy for admitting task-relevant information, thereby reducing interference between tasks; And (iii) multiple specialized attention modules enable solving more complex tasks that cannot be solved by a single specialized function. Combining intuitions from both biological and theoretical findings (Appendix A), we hypothesize that focusing on the information relevant to the current task can facilitate good TP and WP, and consequently systemic generalization by filtering out extraneous or interfering information. In the following section, we describe in detail how we mitigate interference between tasks through task attention.

### 3.2 Preliminary

Continual learning typically involves sequential tasks $t \in \{1, 2, .., T\}$ and classes $j \in \{1, 2, ..., J\}$ per task, with data appearing over time. Each task is associated with a task-specific data distribution $(\mathbf{X}_{t,j}, \mathbf{Y}_{t,j}) \in \mathcal{D}_t$. We consider two popular CL scenarios, Class-IL and Task-IL, defined in Definitions 1 and 2, respectively. Our CL model $\Phi_\theta = \{f_\theta, \tau_\theta, \delta_\theta, g_\theta\}$ consists of a backbone network (e.g. ResNet-18) $f_\theta$, a shared attention module $\tau_\theta$, a single expanding head $g_\theta = \{g_\theta^i \mid i \leq t\}$ representing all classes for all tasks, and a set of task projection vectors up to the current task $\delta_\theta = \{\delta_i \mid i \leq t\}$.

Training DNNs sequentially has remained a daunting task since acquiring new information significantly deteriorates the performance of previously learned tasks. Therefore, to better preserve the information from previous tasks, we seek to maintain a memory buffer $\mathcal{D}_m$ that represents all previously seen tasks. We employ reservoir sampling (Algorithm 3) (Vitter, 1985) to update $\mathcal{D}_m$ throughout CL training. At each iteration, we sample a mini-batch from both $\mathcal{D}_t$ and $\mathcal{D}_m$, and update the CL model $\Phi_\theta$ using experience-rehearsal as follows:

$$\mathcal{L}_{er} = \mathop{\mathbb{E}}_{(x_i,y_i)\sim\mathcal{D}_t} [\mathcal{L}_{ce}(\sigma(\Phi_\theta(x_i)), y_i)] + \alpha \mathop{\mathbb{E}}_{(x_k,y_k)\sim\mathcal{D}_m} [\mathcal{L}_{ce}(\sigma(\Phi_\theta(x_k)), y_k)] \tag{1}$$

where $\sigma(.)$ is a softmax function and $\mathcal{L}_{ce}$ is a cross-entropy loss. The learning objective for ER in Equation 1 promotes plasticity through the supervisory signal from $\mathcal{D}_t$ and improves stability through $\mathcal{D}_m$. Therefore, the buffer size ($|D_m|$) is critical to maintaining the right balance between stability and plasticity in the ER. In scenarios where buffer size is limited ($|D_t| \gg |D_m|$) due to memory constraints and/or privacy reasons, repeatedly learning from the constrained buffer leads to overfitting on the buffered samples. Following Arani et al. (2022), we employ an EMA of the weights ($\theta_{EMA}$) of the CL model to enforce consistency in the predictions through $\mathcal{L}_{cr}$ to enable better generalization (Appendix D.4).

### 3.3 SHARED TASK-ATTENTION MODULE

We seek to facilitate good WP and TP by reducing task interference through task attention. Unlike multi-head self-attention in vision transformers, we propose using a shared, compact task-attention module to attend to features important for the current task. The attention module $\tau_\theta = \{\tau^e, \tau^s, \tau^{tp}\}$ consists of a feature encoder $\tau^e$, a feature selector $\tau^s$, and a task classifier $\tau^{tp}$. Specifically, $\tau_\theta$ is a bottleneck architecture with $\tau^e$ represented by a linear layer followed by Sigmoid activation, while $\tau^s$ is represented by another linear layer with Sigmoid activation. To orient attention to the current task, we employ a linear classifier $\tau^{tp}$ that predicts the corresponding task for a given sample.

We denote the output activation of the encoder $f_\theta$ as $z_f \in \mathbb{R}^{b \times N_f}$, $\tau^e$ as $z_e \in \mathbb{R}^{b \times N_e}$, $\tau^s$ as $z_s \in \mathbb{R}^{b \times N_s}$ and that of $\tau^{tp}$ as $z_{tp} \in \mathbb{R}^{b \times N_{tp}}$, where $N_f$, $N_e$, $N_s$, and $N_{tp}$ are the dimensions of the output Euclidean spaces, and $b$ is the batch size. To exploit task-specific features and reduce interference between tasks, we equip the attention module with a learnable task projection vector $\delta_i$ associated with each task. Each $\delta_i \in \mathbb{R}^{1 \times N_e}$ is a lightweight $N_e$-dimensional randomly initialized vector, learnable during the corresponding task training and then fixed for the rest of the CL training. During CL training, for any sample $x \in \mathcal{D}_t \cup \mathcal{D}_m$, the incoming features $z_f$ and the corresponding task projection vector $\delta_t$ are processed by the attention module as follows:

$$z_e = \tau^e(z_f); \quad z_s = \tau^s(z_e \otimes \delta_t); \quad z_{tp} = \tau^{tp}(z_e \otimes \delta_t). \tag{2}$$

The attention module first projects the features onto a common latent space, which is then transformed using a corresponding task projection vector. As each task is associated with a task-specific projection vector, we expect these projection vectors to capture task-specific transformation coefficients. To further encourage task-specificity in task-projection vectors, AGILE entails an auxiliary task classification:

$$\mathcal{L}_{tp} = \mathop{\mathbb{E}}_{(x,y) \sim \mathcal{D}_t} \left[ \mathcal{L}_{ce}(\sigma(z_{tp}), y^t) \right] \tag{3}$$

where $y^t$ is the ground truth of the task label.

### 3.4 NETWORK EXPANSION

As detailed above, the shared attention module has two inputs: the encoder output $z_f$ and the corresponding task projection vector $\delta_i$. As the number of tasks evolves during CL training, we propose to expand our parameter space by adding new task projection vectors commensurately. These projection vectors are sampled from a truncated normal distribution with values outside $[-2, 2]$ and redrawn until they are within the bounds. Thus, in task $t$ there are $\{\delta_i \in 1, 2, .., t\}$ projection vectors. For each sample, AGILE performs as many forward passes through the attention module as the number of seen tasks and generates as many feature importances ($\in \mathbb{R}^{b \times t \times N_s}$) (see Figure 1). To encourage the diversity among these feature importances, we employ a pairwise discrepancy loss as follows:

$$\mathcal{L}_{pd} = -\sum_{i=1}^{t-1} \mathop{\mathbb{E}}_{(x,y) \sim D_t} \|\sigma(z_s^t) - stopgrad(\sigma(z_s^i))\|_1 \tag{4}$$

where $z_s^i$ is a feature importance generated with the help of the task projection vector $\delta_i$. Since there are multiple feature importances, selecting the right feature importance is non-trivial for longer task sequences. Therefore, we propose to expand $g_\theta = \{g_\theta^i\} \forall i \leq t$ with task-specific classifiers. Each $g_\theta^i$ takes corresponding feature importance $z_s^i$ and the encoder output $z_f$ as input and returns predictions for classes belonging to the corresponding task. We concatenate all the outputs from task-specific classifiers and compute the final learning objective as follows:

$$\mathcal{L} = \mathcal{L}_{er} + \beta\mathcal{L}_{cr} + \gamma\mathcal{L}_{tp} + \lambda\mathcal{L}_{pd} \tag{5}$$

Table 1: Comparison of SOTA methods across various CL scenarios. We provide the average top-1 (%) accuracy of all tasks after training. $^{\dagger}$ Results of the single EMA model.

| Buffer | Methods | Seq-CIFAR10 | | Seq-CIFAR100 | | Seq-TinyImageNet | |
|---|---|---|---|---|---|---|---|
| | | Class-IL | Task-IL | Class-IL | Task-IL | Class-IL | Task-IL |
| - | SGD | $19.62_{\pm0.05}$ | $61.02_{\pm3.33}$ | $17.49_{\pm0.28}$ | $40.46_{\pm0.99}$ | $07.92_{\pm0.26}$ | $18.31_{\pm0.68}$ |
| - | Joint | $92.20_{\pm0.15}$ | $98.31_{\pm0.12}$ | $70.56_{\pm0.28}$ | $86.19_{\pm0.43}$ | $59.99_{\pm0.19}$ | $82.04_{\pm0.10}$ |
| - | PNNs | - | $95.13_{\pm0.72}$ | - | $74.01_{\pm1.11}$ | - | $67.84_{\pm0.29}$ |
| 200 | ER | $44.79_{\pm1.86}$ | $91.19_{\pm0.94}$ | $21.40_{\pm0.22}$ | $61.36_{\pm0.35}$ | $8.57_{\pm0.04}$ | $38.17_{\pm2.00}$ |
| | DER++ | $64.88_{\pm1.17}$ | $91.92_{\pm0.60}$ | $29.60_{\pm1.14}$ | $62.49_{\pm1.02}$ | $10.96_{\pm1.17}$ | $40.87_{\pm1.16}$ |
| | CLS-ER$^{\dagger}$ | $61.88_{\pm2.43}$ | $93.59_{\pm0.87}$ | $43.38_{\pm1.06}$ | $72.01_{\pm0.97}$ | $17.68_{\pm1.65}$ | $52.60_{\pm1.56}$ |
| | ER-ACE | $62.08_{\pm1.44}$ | $92.20_{\pm0.57}$ | $35.17_{\pm1.17}$ | $63.09_{\pm1.23}$ | $11.25_{\pm0.54}$ | $44.17_{\pm1.02}$ |
| | Co$^2$L | $65.57_{\pm1.37}$ | $93.43_{\pm0.78}$ | $31.90_{\pm0.38}$ | $55.02_{\pm0.36}$ | $13.88_{\pm0.40}$ | $42.37_{\pm0.74}$ |
| | GCR | $64.84_{\pm1.63}$ | $90.8_{\pm1.05}$ | $33.69_{\pm1.40}$ | $64.24_{\pm0.83}$ | $13.05_{\pm0.91}$ | $42.11_{\pm1.01}$ |
| | DRI | $65.16_{\pm1.13}$ | $92.87_{\pm0.71}$ | - | - | $17.58_{\pm1.24}$ | $44.28_{\pm1.37}$ |
| | AGILE | $\mathbf{69.37}_{\pm0.40}$ | $\mathbf{94.25}_{\pm0.42}$ | $\mathbf{45.73}_{\pm0.15}$ | $\mathbf{74.37}_{\pm0.34}$ | $\mathbf{20.19}_{\pm1.65}$ | $\mathbf{53.47}_{\pm1.60}$ |
| 500 | ER | $57.74_{\pm0.27}$ | $93.61_{\pm0.27}$ | $28.02_{\pm0.31}$ | $68.23_{\pm0.17}$ | $9.99_{\pm0.29}$ | $48.64_{\pm0.46}$ |
| | DER++ | $72.70_{\pm1.36}$ | $93.88_{\pm0.50}$ | $41.40_{\pm0.96}$ | $70.61_{\pm0.08}$ | $19.38_{\pm1.41}$ | $51.91_{\pm0.68}$ |
| | CLS-ER$^{\dagger}$ | $70.40_{\pm1.21}$ | $94.35_{\pm0.38}$ | $49.97_{\pm0.78}$ | $76.37_{\pm0.12}$ | $24.97_{\pm0.80}$ | $61.57_{\pm0.63}$ |
| | ER-ACE | $68.45_{\pm1.78}$ | $93.47_{\pm1.00}$ | $40.67_{\pm0.06}$ | $66.45_{\pm0.71}$ | $17.73_{\pm0.56}$ | $49.99_{\pm1.51}$ |
| | Co$^2$L | $74.26_{\pm0.77}$ | $\mathbf{95.90}_{\pm0.26}$ | $39.21_{\pm0.39}$ | $62.98_{\pm0.58}$ | $20.12_{\pm0.42}$ | $53.04_{\pm0.69}$ |
| | GCR | $74.69_{\pm0.85}$ | $94.44_{\pm0.32}$ | $45.91_{\pm1.30}$ | $71.64_{\pm2.10}$ | $19.66_{\pm0.68}$ | $52.99_{\pm0.89}$ |
| | DRI | $72.78_{\pm1.44}$ | $93.85_{\pm0.46}$ | - | - | $22.63_{\pm0.81}$ | $52.89_{\pm0.60}$ |
| | AGILE | $\mathbf{75.69}_{\pm0.62}$ | $95.51_{\pm0.32}$ | $\mathbf{52.65}_{\pm0.93}$ | $\mathbf{78.21}_{\pm0.15}$ | $\mathbf{29.30}_{\pm0.53}$ | $\mathbf{64.74}_{\pm0.56}$ |

where $\beta$, $\gamma$, and $\lambda$ are all hyperparameters. At the end of each task, we freeze the learned task projection vector and its corresponding classifier. Figure 1 depicts our proposed approach, which is detailed in Algorithms 1 and 2.

## 4 Experimental results

Table 1 presents a comparison of AGILE with recent rehearsal-based approaches in Class-IL and Task-IL scenarios. The associated forgetting analysis can be found in Appendix C.1. Several observations can be made from these results: (1) Across almost all datasets and buffer sizes, AGILE outperforms the rehearsal-based approaches by a large margin, signaling the importance of task attention in CL. (2) Approaches that employ consistency regularization (e.g., DER++ and CLS-ER) perform considerably better than other approaches. However, as is evident in AGILE, regularization alone is not sufficient to discriminate classes from different tasks. (3) Although approaches aimed at reducing representation drift (e.g., Co$^2$L and ER-ACE) work reasonably well in simpler datasets, they fail to perform well in challenging datasets. For example, in Seq-TinyImageNet where the buffer-to-class ratio is small, their performance is far behind that of AGILE. As shared task attention is largely dependent on task projection vectors to infer task distribution, we contend that fixing task projection vectors after corresponding task training largely limits the representation drift in AGILE. (4) Approaches aimed at improving the quality or quantity of buffered samples (e.g., GCR and DRI) indeed improve over vanilla ER. However, the additional computational overhead in selecting or generating buffered samples can be a problem on resource-constrained devices. On the other hand, AGILE entails compact task attention with task projection vectors and outperforms rehearsal-based approaches by a large margin with little memory and computational overhead.

The task-specific learning approaches, either within a fixed model capacity or by growing, entail parameter isolation to reduce task interference in CL. Similarly, AGILE encompasses task projection vectors to reduce interference between tasks. Figure 2 presents a comparison of AGILE with fixed capacity models (NISPA, CLNP) and growing architectures (PNN, PAE, PackNet, and CPG) trained on Seq-CIFAR100 with 20 tasks (buffer size 500 for AGILE). Across 20 tasks at the end of CL training, AGILE achieves an average of $83.94\%$ outperforming the baselines by a large margin. In terms of parameter growth, PNN grows excessively, while CPG grows by 1.5x, and PAE by 2x. On the other hand, AGILE grows marginally by 1.01x, that too for 20 tasks without compromising the performance in longer task sequences (Table 3).

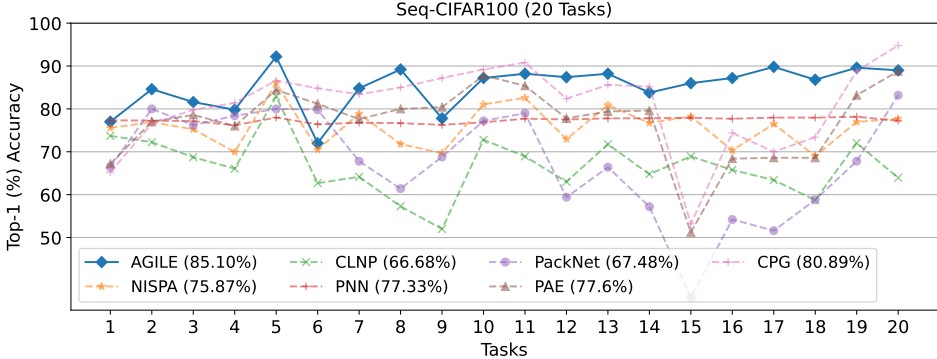

Figure 2: Comparison of AGILE with task-specific learning approaches in Task-IL setting. We report the accuracy on all tasks at the end of CL training with an average across all tasks in the legend. AGILE outperforms other baselines with little memory overhead.

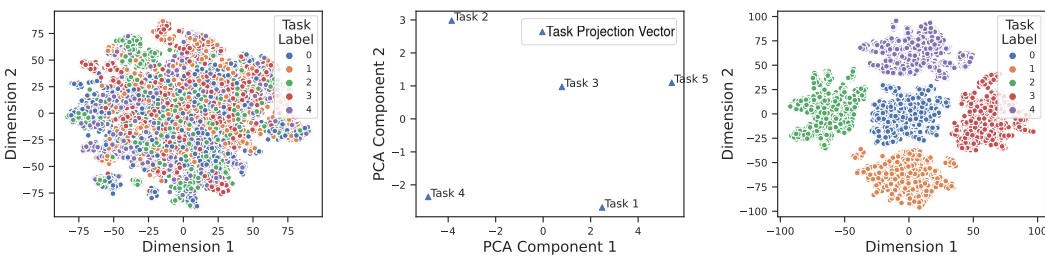

Figure 3: Latent features and task projection vectors after training on Seq-CIFAR100 with 5 tasks. (Left) t-SNE visualization of the latent features of the shared task attention module in the absence of task projection vectors; (Middle) Task projection vectors along leading principle components. (Right) t-SNE visualization of latent features of the shared task attention module in the presence of task projection vectors. Task projection vectors specialize in transforming the latent representations of shared task-attention module towards the task distribution, thereby reducing interference.

### 4.1 HOW AGILE FACILITATES A GOOD WP AND TP?

Figure 3 (left) shows the visualization of t-distributed stochastic neighbor embedding (t-SNE) of latent features in the absence of task projection vectors. As can be seen, samples belonging to different tasks are distributed across the representation space. On the other hand, Figure 3 (right) shows a t-SNE visualization of well-clustered latent features in the presence of task projection vectors. For each sample, we visualize its latent features in task attention after transforming it with the corresponding task projection vector. We also show how task projection vectors are distributed along the principal components using PCA in Figure 3 (middle). AGILE entails a shared task-attention module and as many lightweight, learnable task projection vectors as the number of tasks. As each task projection vector learns the task-specific transformation, they project samples belonging to the corresponding task differently, resulting in less interference and improved WP and TP in CL.

### 4.2 ABLATION STUDY

We aim to determine the impact of each component of AGILE. As previously mentioned, AGILE utilizes consistency regularization through the use of EMA and a shared task-attention mechanism with a single expanding head. Each of these components brings unique benefits to AGILE: consistency regularization aids in consolidating previous task information in scenarios with low buffer sizes, while EMA functions as an ensemble of task-specific models. Furthermore, EMA provides better stability and acts as an inference model in our method. AGILE employs shared task-attention using task-specific projection vectors, one for each task. As the number of tasks increases, selecting

the appropriate task (projection vector) without task identity becomes increasingly difficult (Aljundi et al., 2017). To address this issue, we implement a single expanding head instead of a single head, where each projection vector is responsible for classes of the corresponding task. Table 2 presents the evaluation of different components in Seq-TinyImageNet (buffer size 500). As shown, AGILE takes advantage of each of these components and improves performance in both Class-IL and Task-IL settings.

Table 2: Comparison of the contributions of each of the components in AGILE. Consistency regularization in the absence of EMA implies consistency regularization by storing past logits.

| CONSISTENCY REGULARIZATION | EMA | SINGLE-EXPANDING HEAD | TASK-ATTENTION | CLASS-IL | TASK-IL |
|---|---|---|---|---|---|
| ✓ | ✓ | ✓ | ✓ | $\mathbf{29.30}_{\pm 0.53}$ | $\mathbf{64.74}_{\pm 0.56}$ |
| ✓ | ✓ | ✓ | ✗ | $25.43_{\pm 1.07}$ | $58.89_{\pm 0.84}$ |
| ✓ | ✓ | ✗ | ✗ | $24.97_{\pm 0.80}$ | $61.57_{\pm 0.63}$ |
| ✓ | ✗ | ✗ | ✗ | $19.38_{\pm 1.41}$ | $51.91_{\pm 0.68}$ |
| ✗ | ✗ | ✗ | ✗ | $9.99_{\pm 0.29}$ | $48.64_{\pm 0.46}$ |

## 4.3 PARAMETER GROWTH

AGILE entails as many task projection vectors as the number of tasks. Therefore, the CL model grows in size as and when it encounters a new task. To this end, we compare the parameter growth in AGILE with respect to the fixed capacity model and the PNNs in Table 3. AGILE encompasses as many lightweight, learnable task projection vectors as the number of tasks, specialized in transforming the latent representations of the shared task-attention module towards the task distribution with negligible memory and computational overhead. Compared to fixed capacity models, which suffer from capacity saturation, AGILE grows marginally in size and facilitates a good within-task and task-id prediction, thereby resulting in superior performance even under longer task sequences. On the other hand, PNNs grow enormously in size, quickly rendering them unscalable in longer task sequences.

Table 3: Growth in the number of parameters (millions) for different number of task sequences in Seq-CIFAR100.

| METHOD | 5 TASKS | 10 TASKS | 20 TASKS |
|---|---|---|---|
| FIXED CAPACITY MODEL (WITH EMA) | 22.461 | 22.461 | 22.461 |
| AGILE | 23.074 | 23.079 | 23.089 |
| PNNs | 297.212 | 874.015 | 2645.054 |

## 5 MODEL CHARACTERISTICS

A broader overview of the characteristics of the model is a necessary precursor for the deployment of CL in the real world. To provide a qualitative analysis, we evaluate the recency bias and model calibration for AGILE and other CL methods trained on Seq-CIFAR100 with a buffer size of 500 in Class-IL scenario.

**Model Calibration.** CL systems are said to be well calibrated when the prediction probabilities reflect the true correctness likelihood. Although DNNs have achieved high accuracy in recent years, their predictions are largely overconfident (Guo et al., 2017), making them less reliable in safety-critical applications. Expected Calibration Error (ECE) provides a good estimate of the reliability of models by gauging the difference in expectation between confidence and accuracy in predictions. Figure 4 (right) shows the comparison of different CL methods using a calibration framework (Küppers et al., 2020). Compared to other baselines, AGILE achieves the lowest ECE value and is considerably well-calibrated. By reducing interference between tasks, AGILE enables informed decision-making, thereby reducing overconfidence in CL.

**Task Recency Bias.** When a CL model learns a new task sequentially, it encounters a few samples of previous tasks while aplenty of the current task, thus skewing the learning towards the recent task (Hou et al., 2019). Ideally, the CL model is expected to have the least recent bias with predictions

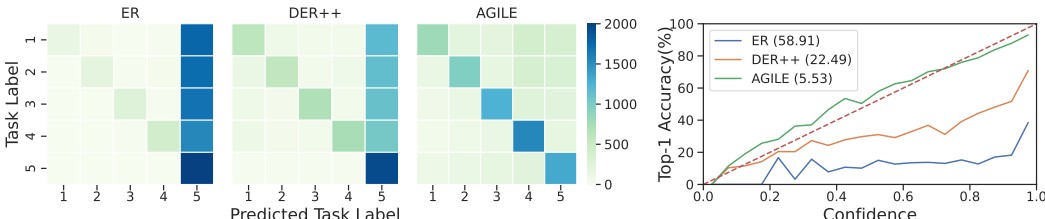

Figure 4: (Left) Confusion matrix of different CL models. ER and DER++ have high recency biases, while AGILE has evenly distributed predictions. (Right) Reliability diagram along with ECE representing model calibration. AGILE is well-calibrated with the lowest ECE value. - - represents the perfect calibration line. All models are trained on Seq-CIFAR100 with 5 tasks.

spread across all tasks evenly. To analyze task-recency bias, we compute the confusion matrix for different CL models .For any test sample, if the model predicts any of the classes within the sample's true task label, it is considered to have predicted the task label accurately. Figure 4 (left) shows that ER and DER++ tend to predict most samples as classes in the most recent task. On the other hand, the predictions of AGILE are evenly distributed on the diagonal. Essentially, AGILE captures task-specific information through separate task projection vectors and reduces interference between tasks, resulting in the least recency bias.

## 6 CONCLUSION

We proposed AGILE, a novel rehearsal-based CL learning approach that employs a compact, shared task-attention module with task-specific projection vectors to effectively reduce task interference in CL. AGILE encompasses as many lightweight, learnable task projection vectors as the number of tasks, specialized in transforming the latent representations of shared task-attention module towards the task distribution with negligible memory and computational overhead. By reducing interference between tasks, AGILE facilitates good within-task and task-id prediction, resulting in superior performance across CL scenarios. With extensive empirical evaluation, we demonstrate that AGILE outperforms the rehearsal-based and parameter-isolation approaches by a large margin, signifying the efficacy of task attention in CL. Extending AGILE to rehearsal-free CL, and exploring different forms of shared task-attention are some of the useful research directions for this work.

## 7 LIMITATIONS AND FUTURE WORK

We proposed AGILE to mitigate task interference and, in turn, facilitate good WP and TP through task attention. AGILE entails shared task attention and as many task projection vectors as the number of tasks. Task projection vectors capture task-specific information and are frozen after corresponding task training. Selection of the right projection vector during inference is nontrivial in longer-task sequences. To address this lacuna, we employ a single expanding head with task-specific classifiers. However, a better alternative can be developed to fully exploit task specificity in the task projection vectors. Second, AGILE strongly assumes no-overlap between classes of two tasks in Class-IL / Task-IL settings. As each task-projection vector captures different information when there is non-overlap between classes, an overlap might create more confusion among projection vectors, resulting in higher forgetting. Furthermore, the shared task-attention module is still prone to forgetting due to the sequential nature of CL training. Therefore, improving task-projection vector selection criterion, extending AGILE to other more complex Class-IL / Task-IL scenarios, and reducing forgetting in shared task-attention module through parameter isolation are some of the useful research directions for this work.

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

# A  THEORETICAL INSIGHT

We consider a widely adopted Class-IL setting within which classes and their domains appear at most in one task, i.e., there is no overlap of classes between tasks.

**Definition 1.** (Class-IL): The CL model encounters $t \in \{1, 2...., T\}$ tasks with $j \in \{1, 2...., J\}$ classes per task sequentially such that the classes belonging to different tasks are disjoint i.e. for task-specific data $(\mathbf{X}_{t,j}, \mathbf{Y}_{t,j}) \in \mathcal{D}_t$, $\mathbf{Y}_{t,j} \cap \mathbf{Y}_{t',j'} = \emptyset$, $\forall j \neq j'$, $\forall t \neq t'$. Given such a setting, the primary goal of the CL model is to learn $\mathbf{P}(y \in \mathbf{Y}_{t,j} \mid \mathcal{D})$.

For any ground event $\mathcal{D}$, Kim et al. (2022) partitioned this probability into two sub-problems, namely within-task prediction (WP) probability: $\mathbf{P}(y \in \mathbf{Y}_{t,j} \mid y \in \mathbf{Y}_t, D)$ and task-id prediction (TP) probability: $\mathbf{P}(y \in \mathbf{Y}_t \mid D)$ as follows:

$$
\begin{aligned}
\mathbf{P}(y \in \mathbf{Y}_{t_0,j_0} \mid D) &= \sum_{t=1,...,n} \mathbf{P}(y \in \mathbf{Y}_{t,j_0} \mid y \in \mathbf{Y}_t, D) \mathbf{P}(y \in \mathbf{Y}_t \mid D) \\
&= \mathbf{P}(y \in \mathbf{Y}_{t_0,j_0} \mid y \in \mathbf{Y}_{t_0}, D) \mathbf{P}(y \in \mathbf{Y}_{t_0} \mid D)
\end{aligned}
\tag{6}
$$

where $t_0$ and $j_0$ represent a particular task and one of its classes, respectively. TP indicates the task-id of the sample, and WP means that the prediction for a test instance is only done within the classes of the task to which the test instance belongs, which is basically the Task-IL problem as follows:

**Definition 2.** (Task-IL): Given the same setting as in Definition 1, the goal of the CL model is to learn the mapping function $f : \mathbf{X} \times \mathbf{T} \to \mathbf{Y}$ i.e. predict the class label $y_{j_0} \in \mathbf{Y}_{t_0}$ for a sample $x$ from task $t_0 \in \mathbf{T}$.

In fact, it is possible to achieve almost zero forgetting in Task-IL (e.g. see (Serra et al., 2018)). However, Class-IL remains challenging due to the difficulty in establishing class boundaries between classes of current and previous tasks. We seek to uncover how Class-IL performance can be further improved. To this end, Let $\mathcal{H}(p,q) = -\sum_i p_i \log q_i$ be the cross-entropy of two probability distributions p and q. We use cross-entropy as a performance measure to assess the relation between WP, TP, and Class-IL. We define the cross-entropy of WP, TP and Class-IL as $\mathcal{H}_{WP}(x) = \mathcal{H}\left(\tilde{y}, \{\mathbf{P}(x \in \mathbf{X}_{t_0,j} \mid x \in \mathbf{X}_{t_0}, D)\}_j\right)$, $\mathcal{H}_{TP}(x) = \mathcal{H}(\bar{y}, \{\mathbf{P}(x \in \mathbf{X}_t \mid D)\}_t)$ and $\mathcal{H}_{Class-IL}(x) = \mathcal{H}\left(y, \{\mathbf{P}(x \in \mathbf{X}_{t,j} \mid D)\}_{t,j}\right)$ respectively, where $\tilde{y}$, $\bar{y}$ and $y$ are ground-truth values $\in \{0, 1\}$. We now describe how WP, TP, and Class-IL are related to each other, and how interference affects their performance.

**Theorem 3.** *If $\mathcal{H}_{WP}(x) \leq \epsilon$ and $\mathcal{H}_{TP}(x) \leq \xi$, then $\mathcal{H}_{Class-IL}(x) \leq \epsilon + \xi$ (Kim et al., 2022).*

For any $\epsilon > 0$ and $\xi > 0$, Theorem 3 establishes a functional relationship between WP, TP, and Class-IL. The theorem states that if $\mathcal{H}_{WP}$ and $\mathcal{H}_{TP}$ are bounded by $\epsilon$ and $\xi$ respectively, then the Class-IL cross-entropy loss $\mathcal{H}_{Class-IL}$ is bounded by the sum of them. Therefore, having good TP and WP, lowers the upper bound of the Class-IL loss.

Task interference arises when multiple tasks share a common observation space but have different learning goals. In the presence of task interference, both WP and TP struggle to find the right class or task, resulting in reduced performance and higher cross-entropy loss. Specifically, in the presence of task interference, the upper bounds of WP and TP increase, indicating the corresponding decrease in performance, i.e. $\mathcal{H}_{WP}(x) \leq \epsilon + \hat{\epsilon}$ and $\mathcal{H}_{TP}(x) \leq \xi + \hat{\xi}$. According to Theorem 3, the upper bound of $\mathcal{H}_{Class-IL}$ will also increase proportionately. Assuming $\hat{\epsilon}, \hat{\xi} \gg 0$, task interference can have a substantial effect on overall Class-IL performance.

Therefore, it is quintessential to reduce task interference in CL to ensure optimum performance. Combining intuitions from both biological and theoretical findings, we hypothesize that focusing on the information relevant to the current task can facilitate good TP and WP, and consequently systemic generalization by filtering out extraneous or interfering information. In the following section, we describe in detail how we mitigate interference between tasks through task attention.

## B  BROADER RELATED WORKS

In section2, we compared and contrasted several methods that are closely related to AGILE. We will now explore the broader related works whose problem statement overlaps with that of AGILE.

Continually learning on a sequence of tasks blurs the decision boundaries between the classes of current task and previous tasks (Lesort et al., 2019). Some approaches address inter-task forgetting indirectly by mitigating the effect of class imbalance in rehearsal-based learning. Attractive and repulsive training (ART) (Choi & Choi, 2022), aims to reduce the correlation between new and old classes through a training strategy that attracts samples from the same class while repelling other similar samples. LUCIR (Hou et al., 2019) proposes a new framework to learn a unified classifier by a combination of cosine normalization, less-forget constraint, and inter-class separation. Although these approaches aim to reduce catastrophic forgetting in CL, they address the fine-grained problem of inter-task class separation without explicitly encouraging a common representation and task-specific learning. In vision transformers, DyToX (Douillard et al., 2022) modified the final multi-head self-attention layer to act as a task attention block using task tokens. However, Dytox is an adhoc approach for transformer architectures and cannot be extended to convolutional architectures.

While rehearsal-based approaches have been highly efficient in CL, repeated learning on a small subset of previous task data results in overfitting thereby inhibiting generalization (Verwimp et al., 2021). Several methods employ augmentation techniques either by combining multiple data points into one (Boschini et al., 2022) or by producing multiple versions of the same buffer sample (Bang et al., 2021). Gradient-based Memory EDiting (GMED) (Jin et al., 2021) proposes to edit individual examples stored in the buffer to create more challenging data for alleviating catastrophic forgetting. Distributionally Robust Optimization (DRO) (Wang et al., 2022b) proposes a principled memory evolution framework to evolve buffer data distribution focusing on population-level and distribution-level evolution. Contrary to the methods that modify memory buffer, Lipschitz Driven Rehearsal (LiDER) (Bonicelli et al., 2022) proposes a regularization objective that induces decision boundary smoothness by enforcing Lipschitz continuity of the model with respect to replay samples. Since these approaches focus on the orthogonal problem of mitigating overfitting in buffer data, they can be integrated into the popular rehearsal-based approaches for further improving generalization in CL.

## C  CHARACTERIZATION OF AGILE

### C.1  FORGETTING ANALYSIS

Learning continuously on a sequence of novel tasks often results in the new information interfering with the consolidated knowledge in the model, causing catastrophic forgetting. Chaudhry et al. (2018) introduced the forgetting measure ($F_T$) to quantify the extent to which previously learned information is retained in the CL model.

Let $a_{ij}$ be the test accuracy of the model for task $j$ after learning task $i$. Then, after training a CL model for $T$ tasks, the forgetting measure $F_T$ for the model is defined as,

$$F_T = \frac{1}{T-1} \sum_{t=0}^{T-1} a_t^* - a_{tT} \tag{7}$$

where $a_t^*$ denotes the best test accuracy for the task $t$. After training for $T$ tasks, $a_t^*$ is typically computed at the task boundaries as,

$$a_t^* = \max_{l \in \{t, t+1, .., T-1\}} a_{lt}, \forall t < T \tag{8}$$

Since the inference model for AGILE is the EMA which is updated stochastically, the maximum accuracy for a previous task could be at any point in the course of training. Therefore, we evaluate AGILE on previous tasks after every epoch to find the maximum accuracy achieved for those tasks to compute the forgetting measure $F_T$. We compare the forgetting measures for different CL methods across different datasets and buffer sizes in the Class-IL setting in Table 4. Evidently, AGILE suffers significantly lower forgetting than other baselines. AGILE encompasses a shared task attention module with task projection vectors that lower the interference between tasks, thereby reducing forgetting.

Table 4: Comparison of the forgetting measure for different CL methods for all scenarios reported in Table 1. Compared to other baselines AGILE suffers the least forgetting.

| BUFFER SIZE | METHODS | SEQ-CIFAR10 | SEQ-CIFAR100 | SEQ-TINYIMAGENET |
|---|---|---|---|---|
| 200 | ER | 61.24±2.62 | 75.54±0.45 | 76.37±0.53 |
| | DER++ | 32.59±2.32 | 68.77±1.72 | 72.74±0.56 |
| | AGILE | **25.40**±0.15 | **22.74**±2.52 | **36.95**±0.51 |
| 500 | ER | 45.35±0.07 | 67.74±1.29 | 75.27±0.17 |
| | DER++ | 22.38±4.41 | 50.99±2.52 | 64.58±2.01 |
| | AGILE | **17.57**±1.45 | **22.71**±0.07 | **23.97**±0.73 |

## C.2 TASK-WISE PERFORMANCE

As CL model learns new tasks in succession, it is exposed to a limited number of examples of earlier tasks, while receiving many more from the task currently being learned. This can cause the model to place more emphasis on recent tasks and less on earlier ones, leading to a bias towards the most recent tasks. Ideally, the CL model is expected to have the least recent bias with predictions spread across all tasks evenly. Figure 5 provides task-wise performance of CL models trained on Seq-CIFAR100 with buffer size 500. As can be seen, the performances of ER and DER++ emanate mostly from the final task, while that of AGILE is much more distributed across tasks.

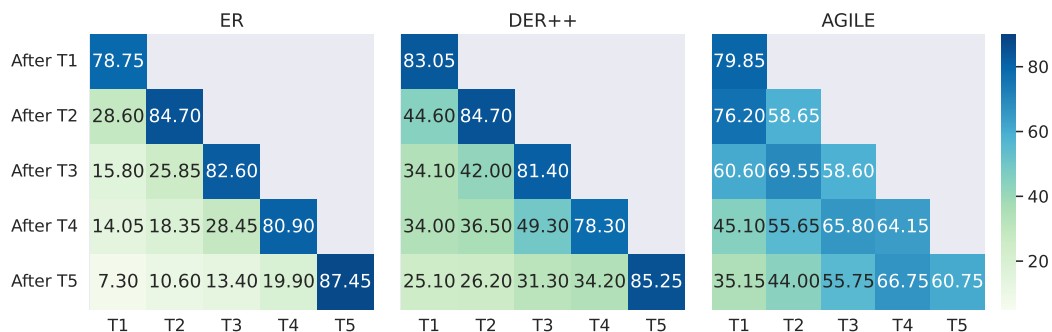

Figure 5: Task-wise performance of CL models trained on Seq-CIFAR100 with buffer size 500. The performances of ER and DER++ mainly emanate from the most recent task, while that of AGILE comes more evenly from all the tasks.

## C.3 STABILITY-PLASTICITY DILEMMA

Stability of a CL model refers to its ability to retain previously learned knowledge, whereas plasticity refers to its ability to adapt to novel information. Every CL model is faced with the dilemma of finding the optimal balance between being stable and being plastic. Consequently, measuring this stability-plasticity trade-off plays a crucial role in analyzing CL models. Sarfraz et al. (2022) proposed a *Trade-off* measure that provides a formal way to estimate the balance between stability and plasticity of the model. If a CL model is trained for $T$ tasks, then the stability ($\mathcal{S}$) of the model is defined as the average performance of all previous tasks, that is,

$$\mathcal{S} = \sum_{t=0}^{T-1} a_{Tt} \tag{9}$$

The plasticity ($\mathcal{P}$) of the model is calculated as the average performance of learning every task for the first time, that is,

$$\mathcal{P} = \sum_{t=0}^{T} a_{tt} \tag{10}$$

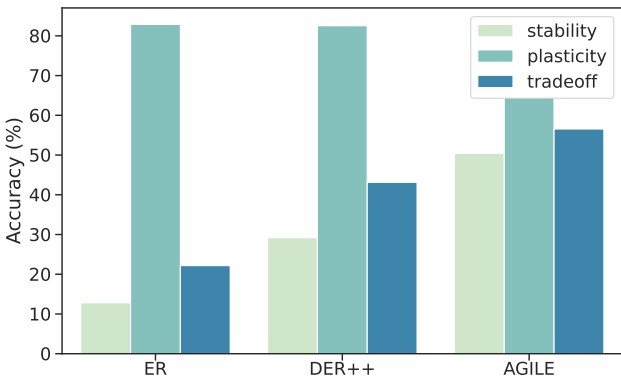

Figure 6: Stability-Plasticity Trade-off for CL models trained on Seq-CIFAR100 with 5 tasks. AG-ILE maintains a better balance between stability and plasticity and achieves the highest trade-off compared to other baselines.

The stability-plasticity trade-off is then measured as the harmonic mean of $\mathcal{S}$ and $\mathcal{P}$.

$$Trade\text{-}off = \frac{2 \times \mathcal{S} \times \mathcal{P}}{\mathcal{S} + \mathcal{P}} \tag{11}$$

Figure 6 compares the stability, plasticity, and trade-off of different CL methods trained on Seq-CIFAR100 with 5 tasks for a buffer size of 500. While ER and DER++ quickly adapt to novel tasks, the new information interferes with the previously learned information leading to low stability. On the other hand, AGILE maintains a better balance between stability and plasticity and achieves a much higher trade-off compared to other baselines.

## D  IMPLEMENTATION DETAILS

### D.1  CONTINUAL LEARNING SETTINGS

We evaluate the effectiveness of our technique in two distinct CL situations: Class Incremental Learning (Class-IL) and Task Incremental Learning (Task-IL). In both Task-IL and Class-IL, each task includes a specific number of new classes that the CL model must learn. A CL model learns multiple tasks, one after the other, while being able to distinguish all the classes it has encountered so far. Task-IL is quite similar to Class-IL, with the only difference being that task labels are also provided during the inference stage, making it the simplest scenario.

### D.2  DATASETS AND MODEL

We obtained the popular CL datasets, Seq-CIFAR10, Seq-CIFAR100, and Seq-TinyImageNet, by dividing the original datasets CIFAR10, CIFAR100, and TinyImageNet into number tasks for the Class-IL and Task-IL scenarios: CIFAR10 into 5 tasks of 2 classes each, CIFAR100 into 5 tasks of 20 classes each, and TinyImageNet into 10 tasks of 20 classes each. In Figure 2, we divide CIFAR100 into 20 tasks of 5 classes each to compare with the parameter isolation methods. To allow for a comprehensive evaluation of different CL methods, we consider two low-buffer regimes 200 and 500, and report average accuracy on all tasks at the end of CL training.

For all of our experiments, we employ ResNet-18 without pre-training as a backbone. Task projection vectors in AGILE are implemented as learnable parameters, while the shared task-attention module is an undercomplete autoencoder-like structure with an additional task-prediction classifier. We emphasize that we use a single expanding head not to be confused with a multiple-head setting. It is to remedy the problem of selecting the right task projection vector during inference. We trained all our models on NVIDIA's GeForce RTX 2080 Ti (11GB). On average, it took around 2

Table 5: Selected hyperparameters for AGILE for all the scenarios reported in Table 1

| DATASET | BUFFER SIZE | EMA PARAMS | | LOSS BALANCING PARAMS | | | | LEARNING RATE | $\delta_i$ DIMENSION |
|---|---|---|---|---|---|---|---|---|---|
| | | $\zeta$ | $\eta$ | $\alpha$ | $\beta$ | $\gamma$ | $\lambda$ | | |
| SEQ-CIFAR10 | 200 | 0.2 | 0.999 | 1 | 0.15 | 1 | 0.1 | 0.07 | 256 |
| | 500 | 0.2 | 0.999 | 1 | 0.10 | 1 | 0.1 | 0.05 | 256 |
| SEQ-CIFAR100 | 200 | 0.05 | 0.999 | 1 | 0.10 | 1 | 0.1 | 0.03 | 256 |
| | 500 | 0.08 | 0.999 | 1 | 0.15 | 1 | 0.1 | 0.07 | 256 |
| SEQ-TINYIMAGENET | 200 | 0.05 | 0.999 | 1 | 0.10 | 1 | 0.1 | 0.05 | 256 |
| | 500 | 0.05 | 0.999 | 1 | 0.10 | 1 | 0.5 | 0.05 | 256 |

hours to train AGILE on Seq-CIFAR10 and Seq-CIFAR100, and approximately 8 hours to train on Seq-TinyImageNet.

### D.3 RESERVOIR SAMPLING

We maintain a fixed size buffer $\mathcal{B}$ following the reservoir sampling strategy (Vitter, 1985). Reservoir sampling samples from a data stream of unknown length by assigning equal probability to each sample to be represented in the memory buffer. Replacements are performed at random once the buffer is full. Algorithm 3 provides the steps to maintain the buffer.

---

**Algorithm 3** Reservoir sampling

1: **Input:** Memory buffer $\mathcal{D}_m$, maximum buffer size $\mathcal{B}$, number of seen samples $N$, current sample $x$, current label $y$
2: **if** $\mathcal{B} > N$ **then**
3:     $\mathcal{D}_m[N] \leftarrow (x, y)$
4: **else**
5:     $k = randomInteger(min = 0, max = N)$
6:     **if** $k < \mathcal{B}$ **then**
7:         $\mathcal{D}_m[k] \leftarrow (x, y)$
8: **return** $\mathcal{D}_m$

---

### D.4 CONSISTENCY REGULARIZATION USING EMA

The CL model's predictions (soft-targets) capture the complex patterns and rich similarity structures in the data. As CL training progresses, soft targets (model predictions) carry more information per training sample than hard targets (ground truths) (Hinton et al., 2015). Therefore, in addition to ground truth labels, soft targets can be leveraged to better preserve the knowledge of the previous tasks. Consistency regularization has traditionally been used to enforce consistency in the predictions either by storing the past predictions in the buffer or by employing an exponential moving average (EMA) of the weights of the CL model. Following Arani et al. (2022), we employ an EMA of the weights of the CL model to enforce consistency in the predictions as follows:

$$\mathcal{L}_{cr} \triangleq \mathbb{E}_{(x_k, y_k) \sim D_m} \|\Phi_{\theta_{EMA}}(x_k) - \Phi_\theta(x_k)\|_F^2 \tag{12}$$

where $\| \cdot \|_F$ is the Frobenius norm, $\Phi_{\theta_{EMA}}$ is the EMA of model $\Phi_\theta$. We update the EMA model as follows:

$$\theta_{EMA} = \begin{cases} \eta \, \theta_{EMA} + (1 - \eta) \, \theta, & \text{if } \zeta \geq \mathcal{U}(0, 1) \\ \theta_{EMA}, & \text{otherwise} \end{cases} \tag{13}$$

where $\eta$ is a decay parameter, $\zeta$ is an update rate, and $\theta$ and $\theta_{EMA}$ are the weights of the CL model and its EMA. As the knowledge of the previous tasks is encoded in the weights of the CL model, we employ EMA for inference instead of the CL model as it serves as a proxy for the self-ensemble of models specialized in different tasks.

## D.5 Hyperparameters

We report the AGILE hyperparameters to reproduce the results reported in Table 1. These hyperparameters were found after tuning with multiple random initializations. In addition to these hyperparameters, we use a standard batch size of 32 and 50 epochs of training for all of our experiments. We use the SGD optimizer and other tools available in PyTorch to build AGILE.

### D.5.1 Hyperparameter Tuning

Table 6: Hyperparamter tuning for AGILE on Seq-CIFAR100 with buffer size 500. As can be seen, AGILE is fairly robust to choice of hyperparameters.

| Varying $\beta$, for $\gamma = 1.0$, $\lambda = 0.1$ | | Varying $\gamma$, for $\beta = 0.15$, $\lambda = 0.15$ | | Varying $\lambda$, for $\beta = 0.15$, $\gamma = 1.0$ | |
|---|---|---|---|---|---|
| $\beta$ | Top-1 Acc % | $\gamma$ | Top-1 Acc % | $\lambda$ | Top-1 Acc % |
| 0.1 | 52.27 | 0.1 | 51.78 | **0.1** | **52.65** |
| **0.15** | **52.65** | 0.5 | 52.46 | 0.2 | 52.33 |
| 0.2 | 51.98 | **1.0** | **52.65** | 0.5 | 52.31 |
| 0.5 | 49.56 | | | | |

