# OpenReview forum: "Mitigating Interference in the Knowledge Continuum through Attention-Guided Incremental Learning"
_ICLR.cc/2024/Conference — Submitted to ICLR 2024_

### Official Review · Reviewer_zq89 · 2023-10-30

**Soundness:** 3 good
**Presentation:** 2 fair
**Contribution:** 2 fair
**Rating:** 5
**Confidence:** 5

**Summary:**

This paper focuses on mitigating task interference in continual learning by introducing a compact task-attention module. It incorporates a set of lightweight, learnable task projection vectors, equal in number to the tasks, which transform the latent representations of a shared task-attention module into task-specific distributions. Additionally, this approach aims to enhance the model's performance in continual learning by jointly addressing the challenges of within-task and task-id prediction.

**Strengths:**

The approach presented in this paper differs significantly from previous methods by combining a task-attention mechanism with minimal memory overhead. It explores the feasibility of reducing interference between tasks and surpasses rehearsal-based approaches in several continual learning scenarios.

**Weaknesses:**

A single lightweight task-specific vector may not be sufficient to adequately represent and distinguish the crucial information among multiple tasks. This approach may not effectively address the issue of catastrophic forgetting.

**Questions:**

1)	This method is less innovative and mainly focuses on solving the task interference problem. How to weigh the importance of solving the interference problem or solving the forgetting problem in continual learning?
2)	The innovation in this paper is that the task-attention module is used to solve the task-id prediction problem, and within-task prediction problem how can it be solved efficiently?
3)	As the number of tasks continues to grow, is there any interference or conflict between these lightweight task-specific vectors?
4)	Can this method be used in other continual learning scenarios, such as Task- free scenario?
5)	Please provide attention-guided visualization experiments showing what the task-specific vector makes the model pay attention to.
6)	In section 3.4 only the extension of the classifiers was carried out, what exactly does the network extension refer to?

---

> ### Author Response · Authors · 2023-11-21
> **Rely to the Reviewer zq89 (1/2)**
>
> We thank the reviewer for their valuable insights on our paper. We provide the responses to the raised concerns below
>
> > A single lightweight task-specific vector may not be sufficient to adequately represent and distinguish the crucial information among multiple tasks.
>
> We appreciate the reviewer's perspective, but respectfully disagree regarding the sufficiency of a single lightweight task-specific vector. For every continual learning approach, mitigating catastrophic forgetting is a balancing act between several factors like model capacity, parameter growth, buffer size and so on. While we acknowledge that a solitary lightweight task-specific vector may not completely eliminate catastrophic forgetting, our innovative attention mechanism has demonstrated significant effectiveness, leading to performance improvements across various continual learning scenarios.
>
> AGILE proposes a novel mechanism to integrate task attention and task specificity with minimal computation and memory overhead so that the model is scalable to longer task sequences. We believe AGILE proves to be a valuable step in the quest of zero forgetting in Class-IL by tackling the problem of task interference.
>
> >This method is less innovative and mainly focuses on solving the task interference problem. How to weigh the importance of solving the interference problem or solving the forgetting problem in continual learning?
>
> We respectfully disagree with the reviewer's assertion that addressing the task interference problem does not contribute to resolving the forgetting issue in continual learning. In the context of continual learning, task interference and forgetting are not mutually exclusive challenges. Inter-task separation is a significant hurdle in Class-IL learning, particularly in realistic scenarios where task-id is unavailable during inference, leading to catastrophic forgetting [1].
>
> The sequential learning of tasks with limited buffer data tends to blur the boundaries between classes distributed across different tasks. Therefore, minimizing task interference becomes crucial for the model to accurately predict the correct class within the current task, free from the influence of classes from other tasks. This approach aids in mitigating the impact of forgetting in continual learning scenarios.
>
> > The innovation in this paper is that the task-attention module is used to solve the task-id prediction problem, and within-task prediction problem how can it be solved efficiently?
>
> AGILE takes a distinctive approach by not explicitly focusing on task-id prediction. Instead, the task projection vectors play a crucial role in transforming feature vectors with task-specific information. This transformation empowers task-specific classifiers to enhance within-task class predictions. Unlike traditional methods that rely on explicit task-id prediction, AGILE leverages these task projection vectors to navigate and capture essential task-specific nuances, contributing to improved model performance within each task.
>
> Furthermore, our novel attention mechanism is a key component designed to mitigate inter-task interference. By incorporating attention mechanisms, AGILE minimizes the impact of information from one task bleeding into another, thus reducing interference between tasks. What sets AGILE apart is its ability to achieve this with minimal memory and computational overhead, making it scalable and efficient for handling longer sequences of tasks in continual learning scenarios. This dual strategy of leveraging task projection vectors and employing a novel attention mechanism positions AGILE as a robust solution for addressing both within-task class prediction and inter-task interference in continual learning.
>
> In section 4.1, we discuss how AGILE facilitates learning a good task-prediction and within-task prediction. We also show how task projection vectors are distributed along the principal components using PCA in Figure 3 (middle). AGILE entails a shared task-attention module and as many lightweight, learnable task projection vectors as the number of tasks. As each task projection vector learns the task-specific transformation, they project samples belonging to the corresponding task differently, resulting in less interference and improved WP and TP in CL.
>
>
>
> [1] Timothee Lesort, Andrei Stoian, and David Filliat. Regularization shortcomings for continual learn- ´ ing. arXiv preprint arXiv:1912.03049, 2019.

---

> > ### Author Response · Authors · 2023-11-21
> > **Rely to the Reviewer zq89 (2/2)**
> >
> > > As the number of tasks continues to grow, is there any interference or conflict between these lightweight task-specific vectors?
> >
> > Once a task is learned, each projection vector is intentionally frozen in AGILE. This deliberate freezing of projection vectors ensures that, for every subsequent creation of a new projection vector, the pairwise discrepancy loss explicitly enforces dissimilarity with respect to the frozen vectors. In essence, this mechanism is designed to prevent overlap and minimize interference or conflict between the newly introduced task-specific projection vectors and those associated with previously learned tasks. The use of pairwise discrepancy loss serves as a regularization technique, maintaining a distinctiveness among projection vectors for different tasks. This approach is instrumental in preserving the unique characteristics of each task and contributes to the model's ability to adapt to new tasks without compromising the knowledge acquired from prior tasks. In summary, the deliberate freezing and distinctiveness enforcement through pairwise discrepancy loss ensure a coherent and interference-minimized integration of task-specific information in AGILE across a reasonable number of tasks.
> >
> >
> > > Can this method be used in other continual learning scenarios, such as Task- free scenario?
> >
> > AGILE is well suited for Class-IL and Task -IL continual learning scenarios. AGILE in its current formulation cannot be applied to general continual learning scenario where the task boundaries are blurry in training and inference. We will include it as a limitation in our final revision.
> >
> > > Please provide attention-guided visualization experiments showing what the task-specific vector makes the model pay attention to.
> >
> >  Figure 3 (Please open the manuscript with a PDF viewer for the figure to load properly)  shows how the latent representation space is transformed by  task-specific vectors. WIthout the task projection the latent space does not show any clear patterns for samples from different tasks. However adding the task-projection vectors transforms the latent space to task specific feature space. These transformed features are then processed by the task specific classifiers to get the final prediction.
> >
> > We thank the reviewer for suggestion on adding visualization on where model is paying attention. In our final revision, we will explore possibilities to include attention visualization showing what the task-specific vector makes the model pay attention to.
> >
> > > In section 3.4 only the extension of the classifiers was carried out, what exactly does the network extension refer to?
> >
> > We regret any confusion caused in the manuscript. Network expansion is intended to show how the model expands when a new task is encountered. AGILE adds a new task projection vector and extends the classification head to accommodate the new classes added.
> >
> > We have thoroughly addressed all concerns raised by the reviewer. If any lingering issues persist, we kindly ask you to bring them to our attention. In the absence of further inquiries, we respectfully request the reviewer to reconsider the assigned score representing the improved confidence in our paper.

---

> > > ### Comment · Area_Chair_mryB · 2023-11-23
> > >
> > > Dear Reviewer,
> > >
> > > The author has provided responses to your questions and concerns. Could you please read their responses and ask any follow-up questions, if any?
> > >
> > > Thank you!

---

### Official Review · Reviewer_2y3j · 2023-10-30

**Soundness:** 1 poor
**Presentation:** 2 fair
**Contribution:** 2 fair
**Rating:** 5
**Confidence:** 4

**Summary:**

Inspired by the notion that most methods that work in a task-incremental scenario can achieve almost zero forgetting, the authors introduce AGILE (Attention-Guided Incremental Learning). The main idea is to break down a class incremental problem into two sub-problems: Task-ID prediction (TP) and within-task prediction (WP). Once the first one is solved, the problem can be treated as a Task-Incremental, as the predicted task-id is already available. The authors suggest using task-specific projections to condition the feature vector. This conditioned vector passes through a task-specific module: task prediction and feature importance. During inference, the output of each module is concatenated to obtain the prediction. The authors demonstrate good performance in both task and class incremental scenarios.

**Strengths:**

- The authors work under the assumption that the incremental Class problem can be transformed into a task-incremental problem.
    - However, I can't entirely agree that this is a "necessary and sufficient" solution. In fact, there is a probability that working the problem in this way helps the model lose generalization in the representations it generates, and the only reason why this does not happen in the proposed solution is that they use a buffer to store previous tasks.
    - Even so it is a problem that is not widely attacked, but that can be a good option in many cases, especially if it's motivated by the idea of GWT.
- The approach comprises many different components that have a good synergy between them. It is beneficial that the authors add Table 2 to show the importance of each loss.

**Weaknesses:**

- Using EMA is a critical point in the proposal, and the authors do not mention it too much. EMA can also be used to reduce weight modification, meaning that it can mitigate forgetting with a favorable beta. The authors present it to increase generalization.
    - Experiments showing evidence that it increases generalization could help mitigate the doubts.
    - Did you have an analysis of the beta value?
- It is challenging to understand where there are linear layers and where there is soft attention in the proposed methods. The image does not help.
    - It could be helpful to decrease the amount of terms, names or losses used in the explanation.
    - For example, from the Figure, one can assume that there is one Task-Attention Module for each task. However, the Task-Attention Module is shared, no?
- Didn’t find Definition 1 and 2.

**Questions:**

- Is EMA used in every method for Table 1? Or just AGILE?
- How much overhead in terms of time is added when adding a Task-Attention Module?
    - Even if the Task-Attention module is shared, it must still be used independently for each task.
- Are you familiar with the work called Bias Correction (BiC) in Continual Learning?
    - There are some similarities that you can find interesting.
    - I don’t remember if it works in class or task-incremental, but there have been extensions that work in class-incremental settings.
- Do you know how your proposal scales with the memory size? I have seen methods that scale well (such as DER), but others could be better (like iCarl).
- Have you tried this approach with a fixed pre-trained model?

---

> ### Author Response · Authors · 2023-11-21
> **Reply to Reviewer 2y3j (1/2)**
>
> We thank the reviewer for their insightful feedback. We provide our responses to the raised concerns below.
> > Using EMA is a critical point in the proposal, and the authors present it to increase generalization.
> Experiments showing evidence that it increases generalization could help mitigate the doubts.
>
> Following CLS-ER[1], we employ EMA as it functions as a self-ensemble of the intermediate model states that leads to better internal representations. Figure 3 in CLS-ER [1] shows that learning with EMA results in convergence to flatter minima, indicating better generalization [2].
> While EMA helps in better generalization, the inter task inference problem still prevails and hinders the performance of the model. On the other hand, AGILE effectively reduces inter task interference with our novel task attention mechanism. As can be seen in Table 1, we outperform CLS-ER by significant margins across all datasets in Class-IL and Task-IL settings.
>
> >Did you have an analysis of the beta value?
>
> We have updated the manuscript with the Hyperparameter Tuning results in the Appendix (section D.5.1). The performance of our model reduces as the $\beta$ value is increased. Higher the $\beta$, higher the restriction on the model to preserve old knowledge, thus limiting its ability to learn new information.
> >It is challenging to understand where there are linear layers and where there is soft attention
>
> Section 3.3 describes the different components in the task attention module in detail. As mentioned there, both feature encoder and feature selector in task attention module is composed of a linear layer followed by Sigmoid activation.
> > It could be helpful to decrease the amount of terms, names or losses used in the explanation.
>
> We regret any confusion caused due to the explanations provided. We have taken utmost care to include all the components and losses in the model so that we are fully transparent. Figure 1, reflects notations used in Algorithm 1 and Proposed Method Section for proper correspondence. Furthermore, we provide the code to the reviewers to ensure complete transparency.
>
> >However, the Task-Attention Module is shared, no?
>
> AGILE uses a single task attention module that is shared across multiple tasks, however it requires as many forward passes through the module as the number of tasks (mentioned in Section 3.4). In Figure 1, we show separate task projection vector and task classifiers for each task whereas the same attention module is shared for each forward pass.
> > Didn’t find Definition 1 and 2.
>
> Owing to the space limitations, we have provided Definitions 1 and 2 in the Theoretical Insights section in the Appendix.
> > is EMA used in every method for Table 1? Or just AGILE?
>
> Among the baseline approaches, CLS-ER is the only approach that uses EMA in its formulation.
> > How much overhead in terms of time is added when adding a Task-Attention Module?
>
> Below we provide the relative time taken to train on Seq-CIFAR100 for 5 tasks with a buffer size of 200. As can be seen, even with as many forward passes through the attention module as the number of tasks, the additional computational overhead incurred in AGILE is very minimal.
>
> Method|ER|DER++|AGILE
> -|-|-|-
> Time|1x|1.5x|1.7x
> >Are you familiar with the work called Bias Correction (BiC) in Continual Learning?
>
> BiC attempts to correct the recency bias in continual learning by adapting the output logits for recent classes with two bias parameters. However, BiC does not entail any task specific learning to reduce task interference in continual learning. Task interference is a significant problem in Class IL that lead to catastrophic forgetting. AGILE, on the other hand, tackles the problem of task interference through the novel task attention mechanism.
>
>
> [1] Elahe Arani, Fahad Sarfraz, and Bahram Zonooz. Learning fast, learning slow: A general continual learning method based on complementary learning system. In International Conference on Learning Representations, 2022.
>
> [2] Pratik Chaudhari, Anna Choromanska, Stefano Soatto, Yann LeCun, Carlo Baldassi, Christian Borgs, Jennifer Chayes, Levent Sagun, and Riccardo Zecchina. Entropy-sgd: Biasing gradient descent into wide valleys

---

> ### Author Response · Authors · 2023-11-21
> **Replying to Reviewer 2y3j (2/2)**
>
> > Do you know how your proposal scales with the memory size? I have seen methods that scale well (such as DER), but others could be better (like iCarl), the exemplar size is also scaled accordingly
>
> Increasing the memory size largely reduces the catastrophic forgetting problem as you have more exemplars from old classes to closely represent old data distribution. This enables the model to learn the decision boundaries better. We expect the margins between rehearsal based approaches to be minimal with large memory size.
> We studied AGILE in a much harder scenario of relatively low memory size regimes where the catastrophic forgetting is exacerbated by task interference. For example, in Seq-TinyImageNet with a buffer size of 200, there are hardly 2 examples per class in the buffer for the model to preserve learnt information. Our novel task attention incorporates task specific learning into the model with task projection vectors and outperforms all other baselines by significant margin.
> > Have you tried this approach with a fixed pre-trained model?
>
> We have not studied this approach with a fixed pre-trained model. We followed the same settings in which the baselines have been studied by training the model from scratch for different datasets.
>
> We thank the reviewer once again for their valuable insights. We hope that we have addressed all the raised concerns sufficiently. We are happy to address any remaining concerns the reviewer may have. In the absence of further questions, we respectfully request the reviewer to reconsider the assigned score, considering the improved confidence in our paper.

---

> > ### Comment · Area_Chair_mryB · 2023-11-23
> >
> > Dear Reviewer,
> >
> > The author has provided responses to your questions and concerns. Could you please read their responses and ask any follow-up questions, if any?
> >
> > Thank you!

---

### Official Review · Reviewer_5d1d · 2023-11-01

**Soundness:** 3 good
**Presentation:** 2 fair
**Contribution:** 2 fair
**Rating:** 5
**Confidence:** 3

**Summary:**

The paper proposes a replay-based CL method utilizing a lightweight task attention module. The module receives features from the feature extractor and performs task-id prediction using the projection vectors for each task. This approach aligns with the findings of a prior theoretical study. The authors conduct comprehensive experiments to demonstrate the benefits of their approach compared to existing baselines and show the effectiveness of the proposed techniques.

**Strengths:**

1. The proposed approach is grounded in a theoretical study.
2. The proposed method outperforms the baselines.

**Weaknesses:**

1. I feel like the paper is written in a rush. The experiment setup is not mentioned in the main paper. It's not clear how many tasks are used in the sequential data (e.g., Seq-CIFAR100), and what architecture is used. I couldn't find where I can find the information in the main text.
2. It's not clear why the shared task-attention module improves WP and TP when this module itself also suffers from forgetting.
3. I couldn't fully understand why this method is better than the existing task-id prediction methods. [1] also builds a task-id prediction module on top of the feature extractor. A more comprehensive and detailed discussion should be included.

Overall, I think this approach is promising, but needs some improvements.

[1] Conditional channel gated networks for task-aware continual learning

**Questions:**

1. How does the model make the final class prediction? Does it first predict the task-id using the attention module and make a within-task prediction?
2. What's the purpose of using the task projection vectors and why is it used to compute both z_s and z_tp?

---

> ### Author Response · Authors · 2023-11-21
> **Reply to Reviewer 5d1d (1/2)**
>
> We express our gratitude to the reviewer for dedicating time to evaluate our paper. Our responses to the raised concerns are presented below.
>
> > I feel like the paper is written in a rush. The experiment setup is not mentioned in the main paper. It's not clear how many tasks are used in the sequential data (e.g., Seq-CIFAR100), and what architecture is used. I couldn't find where I can find the information in the main text.
>
> Owing to space limitations in the paper, we relocated details regarding the experimental setup and datasets to Appendix D. The decision was based on the assumption that, given the adherence to standard model architecture setups similar to other baselines, these details could be appropriately placed in the appendix. However, in the final revision, we will include a reference to these specifics in the Experimental Results section (Section 4) for better accessibility.
>
> > It's not clear why the shared task-attention module improves WP and TP when this module itself also suffers from forgetting.
>
> As outlined in the paper, the performance of Class Incremental Learning (Class IL) can be understood as a synergy between TP (task prediction) and WP (within-class prediction). Given the sequential nature of learning in Continual Learning (CL), the model tends to excel in distinguishing between classes within the same task compared to those across different tasks. In AGILE, we emulate this characteristic by employing task projection vectors that transform backbone features into task-specific latent spaces, implicitly encouraging TP. Subsequently, these transformed features undergo processing by task-specific classifiers, addressing WP.
>
> While it is acknowledged that the task-attention module in AGILE is susceptible to forgetting to some extent, our strategy involves capturing task-specific information within the task projection vectors, while task-agnostic information is preserved in the shared task attention module. As shared, generic information is less prone to forgetting, AGILE exhibits improved performance compared to other approaches.
>
> > I couldn't fully understand why this method is better than the existing task-id prediction methods. [1] also builds a task-id prediction module on top of the feature extractor. A more comprehensive and detailed discussion should be included.
>
> We thank the reviewer for relevant citation and appreciate the opportunity to provide a more comprehensive and detailed discussion on task id prediction methods.
>
> In AGILE, our approach diverges by incorporating a task attention mechanism, which utilizes task projection vectors to transform backbone features into task-specific latent spaces. This transformation is designed to implicitly encourage task prediction (TP). Subsequently, these transformed features undergo processing by task-specific classifiers, addressing within-class prediction (WP). This two-step process distinguishes AGILE from methods like [1], which primarily focus on task-id prediction on top of the feature extractor.
>
> Secondly, the relevant citation [1]  prediction relies on a task classifier, which is trained incrementally in a single-head fashion. Notably, the objective in Eq. 3 in [1] shifts the single-head complexities from a class prediction to a task prediction level. Although the end goal focusses on how the shared information can be filtered out for classification for both AGILE and task-id prediction methods, AGILE approaches the problem differently through theoretically grounded hypothesis.
>
> We agree with the reviewer that a comparison with  task-id prediction modules would add more clarity to the readers. We will update our final revision with extensive related works section focussing on task-id prediction methods.

---

> > ### Author Response · Authors · 2023-11-21
> > **Reply to Reviewer 5d1d (2/2)**
> >
> > > How does the model make the final class prediction? Does it first predict the task-id using the attention module and make a within-task prediction?
> >
> > AGILE does not rely on explicit task-id prediction. Instead, it employs task-specific projection vectors for the transformation of feature vectors associated with each task. These transformed vectors are subsequently processed through task-specific classifiers, facilitating within-class predictions. The outputs from these classifiers are then concatenated to form the final output vector, encompassing all classes.
> >
> > > What's the purpose of using the task projection vectors and why is it used to compute both z_s and z_tp?
> >
> > The task projection vector plays a pivotal role in capturing task-specific information. In the encoder module of the shared task attention, features from the backbone are projected into a common latent space. This projection is further transformed using a task projection vector, directing the features to their respective task-specific latent spaces. Subsequently, these transformed features undergo processing through the t_s and t_tp modules to generate the decoded feature vector and facilitate auxiliary task classification learning.
> >
> > We have meticulously addressed all concerns raised by the reviewer. If any lingering issues persist, we kindly request you to bring them to our attention. In the absence of further inquiries, we respectfully urge the reviewer to reconsider the assigned score, taking into account the enhanced confidence in the quality of our paper.

---

> > > ### Comment · Area_Chair_mryB · 2023-11-23
> > >
> > > Dear Reviewer,
> > >
> > > The author has provided responses to your questions and concerns. Could you please read their responses and ask any follow-up questions, if any?
> > >
> > > Thank you!

---

### Official Review · Reviewer_pnnH · 2023-11-01

**Soundness:** 2 fair
**Presentation:** 2 fair
**Contribution:** 2 fair
**Rating:** 5
**Confidence:** 4

**Summary:**

This paper introduces a novel rehearsal based continual learning approach which use a shared task-attention module to mitigate the task interference. The shared task-attention module compresses the task specific information to some trainable parameters.

**Strengths:**

1. The framework achieves fairly good results compared with baselines.
2. The paper is written clearly and easy to follow.

**Weaknesses:**

1. Novelty concern. I would like to point out that the idea of leveraging trainable parameters to store task information has been investigated in previous works [*] [**]. L2P has shown its effectiveness in continual learning areas in recent years.

2. Lack of a comprehensive comparison. There are many works using prompting (learnable parameters) in continual learning and achieving SOTA performance. I suggest the author conduct a comprehensive comparison with these works.

[*] Learning to prompt for continual learning, CVPR 2022.

[**] DualPrompt: Complementary Prompting for Rehearsal-free Continual Learning, ECCV 2022.

**Questions:**

Could the author conduct a comprehensive comparison with CL works using prompting (learnable parameters)?

---

> ### Author Response · Authors · 2023-11-21
> **Reply to Reviewer pnnH**
>
> We thank the reviewer for taking the time to review our paper. We provide our responses to the raised concerns below.
>
> > Novelty concern. I would like to point out that the idea of leveraging trainable parameters to store task information has been investigated in previous works [*] [**]. L2P has shown its effectiveness in continual learning areas in recent years.
>
> While the idea of leveraging trainable parameters has been investigated, we believe the novelty comes from how the idea is implemented. Transformer architectures allow us to somewhat intuitively extend the input tokens with special tokens (say class token) that capture specific information. Learning to prompt and DualPrompt are both transformer based architectures that use special prompt tokens with pre-trained models to then learn task specific information.
>
> However such an extension is not straightforward for CNN architecture. CNN does not have any attention mechanisms or tokens. AGILE leverages parameter isolation to bring in task specificity with little computational or memory overhead. Our novelty comes in emulating such task attention through a shared under complete autoencoder and task projection vectors. These projection vectors then capture task specific information through Our auxiliary task classification loss and pair wise discrepancy loss. Even in CNN architectures, several approaches have tried to capture task specific information through parameter isolation. We compare against such approaches in Figure 2.
>
> In this study, we have explored the integration of our method into the widely used CNN architecture, specifically ResNet18. We have examined various baseline methods within the CNN framework and conducted a comparative analysis, demonstrating that AGILE achieves superior performance due to its innovative task attention mechanism. While AGILE's task attention mechanism utilizes simple undercomplete autoencoders and tensors, devoid of architecture-specific layers like convolutions, we concur with the reviewer's suggestion that AGILE could potentially be generalized to transformers. However, we acknowledge that extending AGILE to different architectures, such as transformers, is a non-trivial task requiring extensive experiments and hyperparameter tuning. Such endeavors go beyond the scope of this rebuttal period. Recognizing the potential benefits of extending AGILE to transformers for enhanced generalizability and comparison with methods such as L2P, DualPrompt, we propose to address this in future work and will update our limitations section accordingly.
>
> > Lack of a comprehensive comparison. There are many works using prompting (learnable parameters) in continual learning and achieving SOTA performance. I suggest the author conduct a comprehensive comparison with these works.
>
> We respectfully disagree with the reviewer's assertion that our comparison with other state-of-the-art (SOTA) approaches is not comprehensive. It is crucial to emphasize that our current focus is confined to CNN architecture. Given the well-established exploration of continual learning within CNN architectures, we have conducted a thorough comparison with various popular and recent rehearsal approaches, as detailed in Table 1 under consistent experimental settings. Additionally, our evaluation extends to the comparison with several methods specifically designed to capture task-specific information, as illustrated in Figure 2.
>
> > Could the author conduct a comprehensive comparison with CL works using prompting (learnable parameters)?
>
> The current scope of our work is primarily centered on CNN architecture. While we have extensively compared our approach with various state-of-the-art rehearsal methods and those focusing on task-specific information capture within the CNN framework, the specific comparison with continual learning works using prompting and learnable parameters falls outside the scope of our study. We acknowledge the importance of this avenue for future exploration and welcome it as a potential direction for further research.
>
> We have thoroughly addressed all concerns raised by the reviewer. If any lingering concerns remain, we kindly ask you to bring them to our attention. In the absence of further questions, we respectfully request the reviewer to reconsider the assigned score, considering the improved confidence in our paper.

---

> > ### Comment · Area_Chair_mryB · 2023-11-23
> >
> > Dear Reviewer,
> >
> > The author has provided responses to your questions and concerns. Could you please read their responses and ask any follow-up questions, if any?
> >
> > Thank you!

---

### Official Review · Reviewer_n4fn · 2023-11-04

**Soundness:** 2 fair
**Presentation:** 3 good
**Contribution:** 2 fair
**Rating:** 3
**Confidence:** 5

**Summary:**

The paper introduces a rehearsal-based method called AGILE to tackle the class-incremental learning setting in continual learning. Specifically, the paper leverages learnable task embedding vectors and shared task-attention module for better mitigating task interference. Experimental results on benchmark datasets demonstrate the effectiveness of the method.

**Strengths:**

- The paper reads well and is easy to follow.
- Class-incremental learning is indeed a more challenging setting than task-incremental learning.

**Weaknesses:**

- The idea of using task-attention or task embedding vector is not quite novel. For example, DyTox [1] also has a task attention module, L2P [2] leverages task-specific prompts.
- Following the first one, I think the paper misses several recent competitive methods to compare against. For example, I understand both DyTox and L2P are based on transformers. However, if the proposed method AGILE is generalizable enough, it should be compatible with transformer architectures as well, making comparison with more advance methods like DyTox, L2P possible.
-
- The contents in middle and right subfigures in figure 3 seems missing?

[1] Douillard, Arthur, et al. "Dytox: Transformers for continual learning with dynamic token expansion." CVPR 2022
[2] Wang, Zifeng, et al. "Learning to prompt for continual learning." CVPR 2022

**Questions:**

- I understand the method is based on rehearsal, what if the rehearsal part is removed. Will the remaining design lead to improvement upon the baselines without rehearsal as well?
- See weaknesses for the rest questions.

---

> ### Author Response · Authors · 2023-11-21
> **Reply to reviewer n4fn**
>
> We thank the reviewer for their valuable feedback and insights. We provide the responses for the raised concerns below.
>
> > The idea of using task-attention or task embedding vector is not quite novel
>
> Transformer architecture provides a rather straightforward way to extend self attention mechanism with learnable parameters to capture any specific information. However such attention learning is not intuitive in other architectures, like CNNs.Our novelty is in emulating the task attention mechanism with a bottleneck architecture and encouraging task specific learning with task projection vectors by transforming the encoded latent space output in CNNs. Our auxiliary task classification loss and pair wise discrepancy loss enables the projection vectors to capture task specific information.
>
> For the scope of this work, we have studied the integration of our method into the widely popular CNN architecture (ResNet18). Several baseline methods are all studied in the CNN architecture, We compare AGILE against these methods under the same experimental setting and show that AGILE can achieve better performance through our novel task attention mechanism.
>
> Since AGILE uses simple under complete auto-encoders and tensors( no architecture specific layer like convolutions ) for our task attention mechanism, we agree with the reviewer that AGILE can be generalized to transformers. However, extending it to other architectures like transformers may not be trivial and it warrants quite some experiments and hyperparameter tuning which is beyond the scope of this rebuttal period. We acknowledge that extending AGILE to transformers could in fact strengthen the method for its generalizability. We leave that as a future work and will update our limitation section to reflect the same.
>
> > The contents in middle and right subfigures in figure 3 seems missing?
>
> We see that the PDF is not loading fully in browser tabs. We request the reviewer to use a PDF viewer application. We will update the PDF in the meantime for proper loading in browser windows.
>
> > I understand the method is based on rehearsal, what if the rehearsal part is removed. Will the remaining design lead to improvement upon the baselines without rehearsal as well?
>
> Removing rehearsal would drastically affect all the rehearsal based approaches including AGILE. All the methods would be reduced to a simple SGD based fine tuning, a naive baseline provided in Table 1. Even though AGILE has task specific projection vectors, the shared task attention module would undergo catastrophic forgetting without the rehearsal and the captured information in projection vectors may cease to be useful.
>
>
> We have diligently addressed all concerns raised by the reviewer. Should any concerns persist, we kindly request that you bring them to our attention. If no further questions arise, we kindly request the reviewer to reconsider the assigned score, taking into account the enhanced confidence in our paper.

---

> > ### Comment · Area_Chair_mryB · 2023-11-23
> >
> > Dear Reviewer,
> >
> > The author has provided responses to your questions and concerns. Could you please read their responses and ask any follow-up questions, if any?
> >
> > Thank you!

---

### Meta-Review · Area_Chair_mryB · 2023-11-29

**Metareview:**

The paper proposes Attention-Guided Incremental Learning' (AGILE) method for continual learning. After taking into account the reviews from all the reviewers, the strengths and weaknesses of the paper are summarized below. As the weakness outweighs the merits, given the current status, AC recommends rejection.

Strengths:

The problem this paper tries to tackle is important to study. It is well-motivated.
The paper introduces some novelties in the method designs, especially the task-attention module.

Weaknesses:

1. The main concern of most reviewers is the lack of comparisons with other continual learning baselines, such as L2P, and DyTox. Discussion about the difference between the proposed method and the baselines is missing.
2. Moreover, the paper seems to be prepared in a rush with unclear text and ambiguous figure designs.
3. There are limited contributions in the method design. For example, The effect of EMA can be analyzed further. The effect of hyperparameters can be explored. How this method can be adapted for free-task-boundary continual learning (currently, the model is unable to do such tasks) can be discussed.

**Justification For Why Not Higher Score:**

see the weakness above (mainly on the limited novelty of the proposed method + poor clarity of the paper + lack of comparison with other methods, such as prompt-based continual learning methods)

**Justification For Why Not Lower Score:**

N/A

---

### Decision · Program_Chairs · 2024-01-16

Reject